# Photovoice in Aggression Management Training for Medical and Nursing Students—A Pilot Study

**DOI:** 10.3390/healthcare12090873

**Published:** 2024-04-23

**Authors:** Jakub Lickiewicz, Laura S. Lorenz, Bettina Kolb

**Affiliations:** 1Department of Health Psychology, Jagiellonian University Medical College, 31-501 Krakow, Poland; 2Heller School for Social Policy and Management, Brandeis University, Waltham, MA 02453, USA; laura@pvww.org; 3Department of Sociology, University of Vienna, 1090 Vienna, Austria; bettina.kolb@univie.ac.at

**Keywords:** photovoice, aggression, aggression management training, medicine students, nursing students

## Abstract

Aggression towards medical staff in the healthcare workplace is a common global concern. Measures to mitigate consequences of patient aggression include training through Aggression Management Programs (AMPs), which have been shown to increase students’ self-efficacy and self-confidence. To encourage better engagement with a 30 h required AMP training, the study piloted an adapted photovoice activity with 58 students of medicine and nursing. Each student took one to three photos depicting their perceptions, feelings, and experiences of patient aggression in the workplace and discussed them in a course session. Their photos showed types of aggression in psychiatric settings, and their consequences for patients and students. Photo strategies included showing ‘actors’ or toy figures in aggressive encounters; tools to control aggression in psychiatric settings (e.g., mechanical restraints and syringes); and symbolic photos showing violence to the heart (emotional impact). Adding photovoice elements to the established AMP training appeared to contribute to student reflection on their individual perspectives on patient aggression in the workplace and help students to link their subjective experiences and theoretical learning. In future, incorporating pre-test and post-test questionnaires measuring empathy, attitudes, or critical thinking could help to decipher any changes in AMP effectiveness due to the use of a self-directed photovoice activity.

## 1. Introduction

Aggression towards medical staff is a common concern [1] that negatively affects professionals and patients [2]. Workplace violence “is defined as aggression when staff members are abused, intimidated, or attacked in circumstances related to their work…” ([2], p. 1). Workplace violence is known to negatively impact medical and nursing students, particularly women [3]. Workplace violence results in feelings of anger, frustration, hopelessness, and self-blame among medical staff, and consequently, a negative attitude toward patients [4]. The consequences of violence in the healthcare workplace include absenteeism, personnel loss, and employee burnout [5]. Verbal aggression (screaming and threats) is the most common, and contact with aggressive patients and their families generates anger and frustration among healthcare staff and a negative attitude toward aggressive patients [6].

In Poland, the frequency of aggression towards healthcare workers has been documented in hospitals (98.6% of employees), psychiatric wards (98.5%), and primary care facilities (90%) [7]. One-third of employee respondents reported experiencing aggression every day, and 31.3% reported experiencing it at least once a week. A study of aggression toward nurses in Poland reported that every nurse respondent had experienced patient aggression, with verbal attacks being the most prevalent [8]. Research conducted among employees of hospitals in Poland’s Lower Silesia showed that all respondents were exposed to aggression while performing their duties. During the 12 months of the study, 96% of respondents saw aggression against another staff member and 91% were victims of patient assault, while 33.3% assessed that they were victims of aggression every day, and 31.3% at least once a week [7].

Measures taken to mitigate the negative consequences of aggression by patients in the healthcare workplace include legislation, incident monitoring, and education and training. Aggression Management Programs (AMPs) focus on conceptual and practical training to understand the mechanism of aggression, its risk factors, and practical advice for dealing with it [9]. AMP training is recommended for building skills to manage aggression and violence [10]. AMP training can have a positive impact on knowledge related to violence de-escalation, confidence to manage aggression, and performance [11], and may have a positive influence on job satisfaction [12].

The AMP Therapeutic Management of Aggression (TERMA) was developed in Norway and has been used in Norway, Sweden, and Poland [3]. Its model has three elements: positive appreciation of patients (moral perception and compassion), emotional regulation (suppression and emotional equilibrium), and effective structure (routine direction and conduct) [13]. TERMA’s goal is to develop knowledge, skills, and proactive behaviours to prevent aggression from occurring. TERMA is a 30 h required program and consists of lectures, seminars, case studies, and role-playing. Topics include the psychological mechanisms of aggressive behaviour, de-escalation methods, legal regulations, and breakaway techniques. A study of TERMA outcomes among 276 medical and nursing students of showed a significant positive impact on all students’ self-efficacy and self-confidence and on the perception of aggression among female students [3].

Studies exploring the effectiveness of AMPs have thus far investigated skill- and confidence-building outcomes but have not investigated outcomes in student attitudes, empathy, or emotional regulation, as called for by TERMA’s model. The training sought to enhance TERMA training outcomes by creating an opportunity for students to gain a more profound understanding of aggression in psychiatric settings, gain deeper insights into their emotions related to patient aggression, and engage in their TERMA learning in a new, more creative way. This paper reports on findings and observations from a pilot activity in which TERMA students used a modified version of photovoice, which asks participants to be visual researchers as they take photos to answer research questions, discuss their pictures in a group, write captions, and present findings to decisionmakers [14].

Photovoice has been used as a teaching tool with undergraduate and graduate nursing and medical students, health science students, and health profession students completing short-term field experiences [15,16,17,18,19]. Studies in seven countries have found that using photovoice in public health, nursing, and medical student education resulted in increased recognition of students’ shared humanity with patients [19], increased ability to express emotions and have empathy for patients [17], better engagement of students in their learning [18], improved practice with observation and interpretation skills [16], improved critical thinking skills [15], and improved application of course concepts [19].

## 2. Materials and Methods

The study took place between January and March 2024. Fifty-eight students of Jagiellonian University Medical College, Krakow, Poland, participated in the pilot program—24 medical students (10 males) and 34 nursing students (2 males) in single-discipline classes (medicine or nursing). A single faculty member facilitated TERMA and the photovoice activity, to avoid any extraneous variables connected with faculty personality or training style. The structured TERMA training was based on the TERMA methodology and program and consisted of 30 training sessions. Students were acquainted with the psychological mechanisms of aggressive behaviour, aggression de-escalation methods, legal regulations, and breakaway techniques. Students learned about the phases of aggressive incidents (pre-phase, during, and post-phase) and the rules for dealing with the negative consequences of aggressive behaviour against medical personnel. In a previous study of the AMP program where TERMA was used, Lickiewicz et al. (2020) observed differences in perceptions of aggression between males and women. After the training, women saw aggressive behaviour as a defence of territory and behaviour that served a particular purpose. The changing perception may have resulted from the better understanding of the motivation and causes of aggression. No such effect was observed in the male group [3].

The photovoice activity was offered near the end of the 30-week training and involved the following: (a) learning about ethical issues related to taking photos for research, e.g., no photos of a patient’s face or without a person’s consent; (b) working individually to take up to three photos of aggression in the workplace; and (c) choosing how to complete the task, e.g., take one photo or create a series of images (narrative story). Students were asked to take one to three photos of “aggressive behaviours in medical care” and discuss them with their classmates and the faculty member. They worked independently, uploaded their work to the university server, and analysed their photos in a course session facilitated by the first author, who used a projector to show each image. The first question posed to the group was, “What can we see here in the photo?” Following the group discussion, each photographer shared what the photo meant to them. The pilot activity was not graded, to avoid competition and encourage creativity.

Once the photovoice activity was complete, students were invited to have their photos anonymously included in a study of the pilot. They were allowed to withdraw from participation in the research by deleting their pictures from the server without explaining their reason and without any impact on their AMP grade; however, no students withdrew their photos. The pilot study’s aims were to (1) identify and compare the issues represented in the students’ photos and captions and (2) assess the method’s effectiveness at enhancing TERMA outcomes by creating an opportunity to discuss the emotional aspects of patient aggression in a psychiatric setting.

Photovoice has no single, established methodology for analysing the participants’ pictures, although the thematic analysis of photos and their related interview or written content is a commonly used approach [20,21]. Thematic analysis methods focus on what is being shown or said. Narrative analysis methods focus on how visuals and texts communicate meaning [22] (Riessman, 2008). In our study, we used narrative analysis methods to identify and describe documentary photos versus symbolic photos, as well as photos in a series [18,22]. Another approach to visual analysis is photo production, or how the participating photographers obtained their photos, including type of camera, time allowed for photo-taking, and image product strategies, for example, taking one photo versus several photos in a series [18]. Our data interpretation focused on describing the images, an essential task, as recommended by Riessman (2008) [22]. Our interpretive work followed the model provided by Dongre (2011) to describe what was observed in the photos, what was interpreted by students, and what was noticed by faculty [16].

The research has the acceptance of the Ethical Committee Jagiellonian Universirty Medical College, Krakow, Poland (no. 118.6120.126.2023).

## 3. Results

The study database has 76 photos: 51 by nursing students and 25 by medical students. Students uploaded three types of photo submissions: single photos that documented aspects of aggression in the healthcare workplace (documentary photos), single photos that showed their feelings about aggression in the healthcare workplace in a symbolic way (symbolic photos), and photos in a series that showed an evolving story (photo narratives).

The photos reveal the different aspects of patient aggression in psychiatric settings including types, consequences, treatment tools, and prevention strategies. Photo-taking production strategies included depicting aggressive encounters using ‘actors’ or lego figures; showing tools to control aggression (hospital beds, restraints, syringes, and medication); and taking symbolic photos to depict the emotional impact of aggression in psychiatric settings, e.g., showing a knife causing harm to the heart. Below, we provide a sampling of nursing student photos taken for the assignment followed by a similar sampling by students of medicine. For each discipline, a documentary photo, a symbolic photo, and a photo series are shown (Figure 1, Figure 2, Figure 3, Figure 4, Figure 5 and Figure 6). For each photo or photo series, a brief excerpt from the photographer’s verbal explanation of the photo’s meaning is provided.

**Nursing students**: During their training, nurses spend time with patients, giving prescribed medications and, at times, facilitating group therapy. Their photos and captions for this pilot activity primarily documented their training experiences in the workplace. Photos by nursing students were more likely to show (a) the tools of their work, (b) staged situations, and (c) consequences for patients. Several showed social media content attacking nurses, another form of aggression they experience. Figure 1 shows a nursing student’s documentary photo and caption.
Figure 1Sample documentary photo and caption by a nursing student participating in the photovoice activity. Mechanical restraints as consequences of aggression.
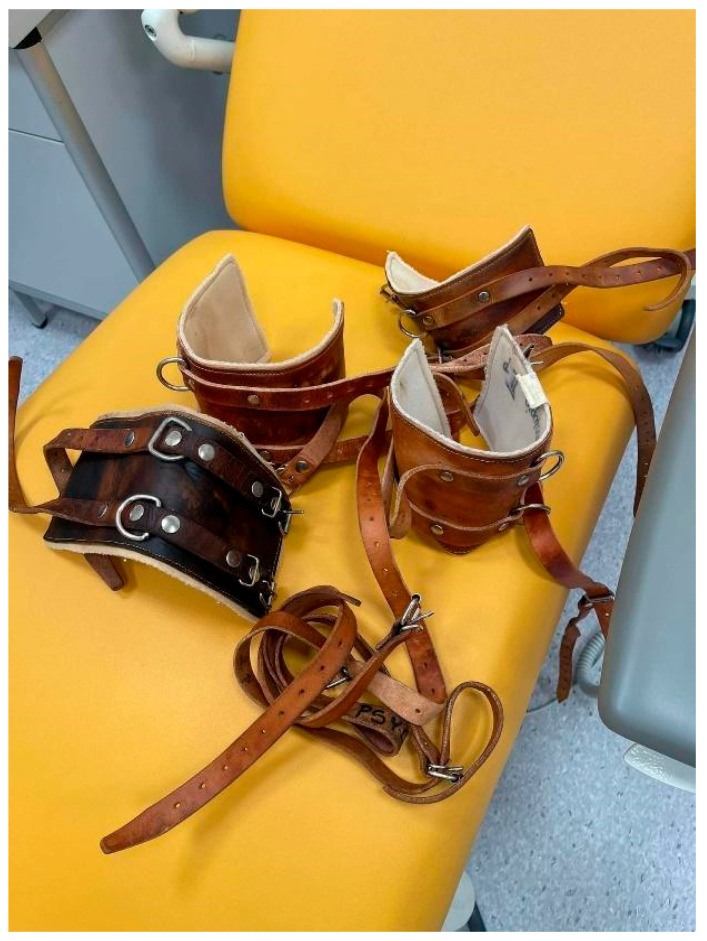


The photo in Figure 1 shows traditional restraints that remain in use in psychiatric settings in Poland. The restraints, placed on a treatment room chair, depict the consequences of workplace aggression for patients.

Now, let us look at a symbolic photo submitted by a nursing student for the photovoice activity. Figure 2 depicts the consequences of aggression on the student. The photos show a partially peeled banana on a blue cloth.
Figure 2Sample symbolic photo and caption by a nursing student participating in the photovoice activity. You are like a banana, but aggression can reveal your interior.
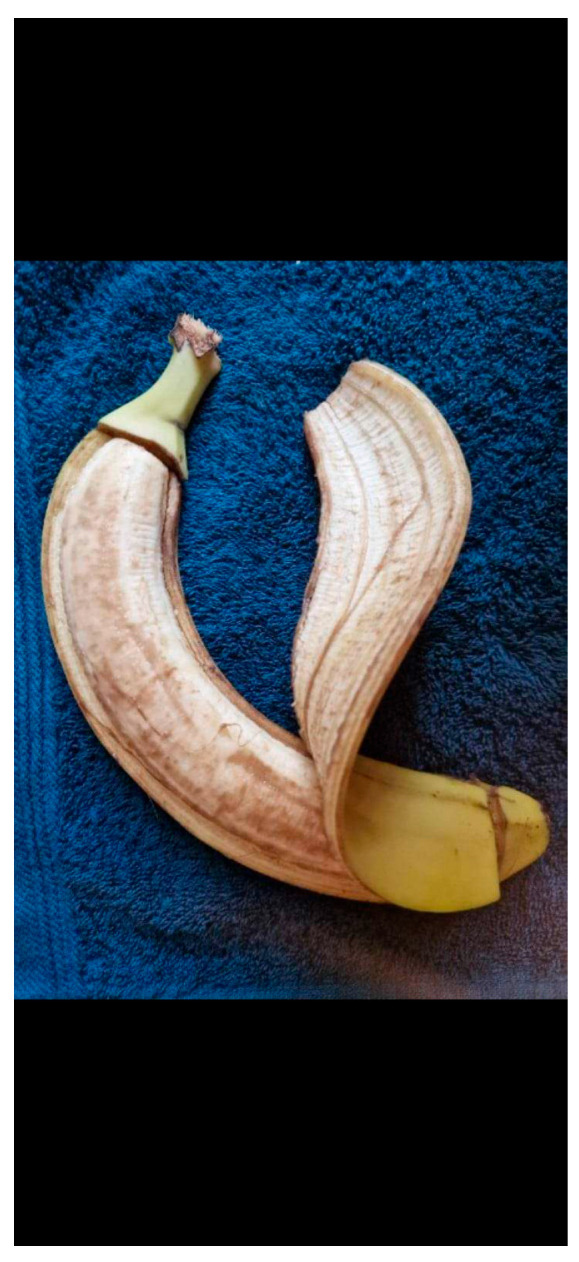


In the photo, the inside of the fruit, normally white and fresh, is brown after exposure to the air. The fruit of a banana with its skin peeled away is a symbolic expression of the emotions engendered by aggression in the workplace. The photo symbolises the impact of aggression on the provider.

Figure 3 is an example of nursing students illustrating a story through a series of photos. Nursing students used the option of submitting a series of three photos to show fellow students or lego figures in staged situations depicting behaviour in the healthcare workplace.
Figure 3Sample photo narrative by a nursing student participating in the photovoice activity. When you support other people, it helps, but aggression is like a virus, it spreads.
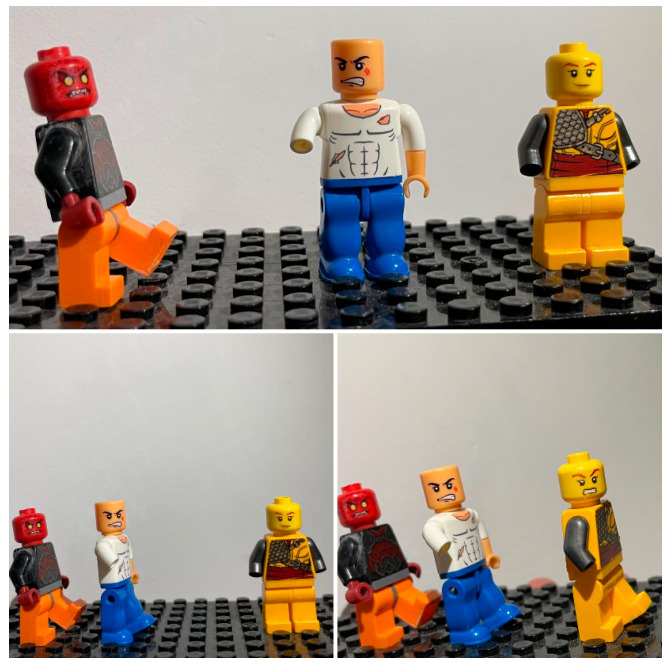


The lego characters in Figure 3 represent patients and students. In the first photo, two figures have aggressive expressions on their faces as they direct angry gazes at each other. In the second photo, they approach the healthcare provider with angry expressions. In the third photo, all three figures have angry expressions, and one figure is directing its gaze at the photo audience.

**Medical students:** Medical students were more likely than nursing students to take symbolic photos and more likely to depict situations in the healthcare setting using photos taken outside the workplace. For example, a photo of a large truck carrying a small truck symbolised the need for support after an incident of patient aggression. Medical students also used drawings and staged situations to depict harm to the heart (aggression from knives, a pick, and a closed fist). Figure 4 shows a medical student’s documentary photo and caption.
Figure 4Sample documentary photo and caption by a student of medicine participating in the photovoice activity. The process of treatment might be broken because of aggression.
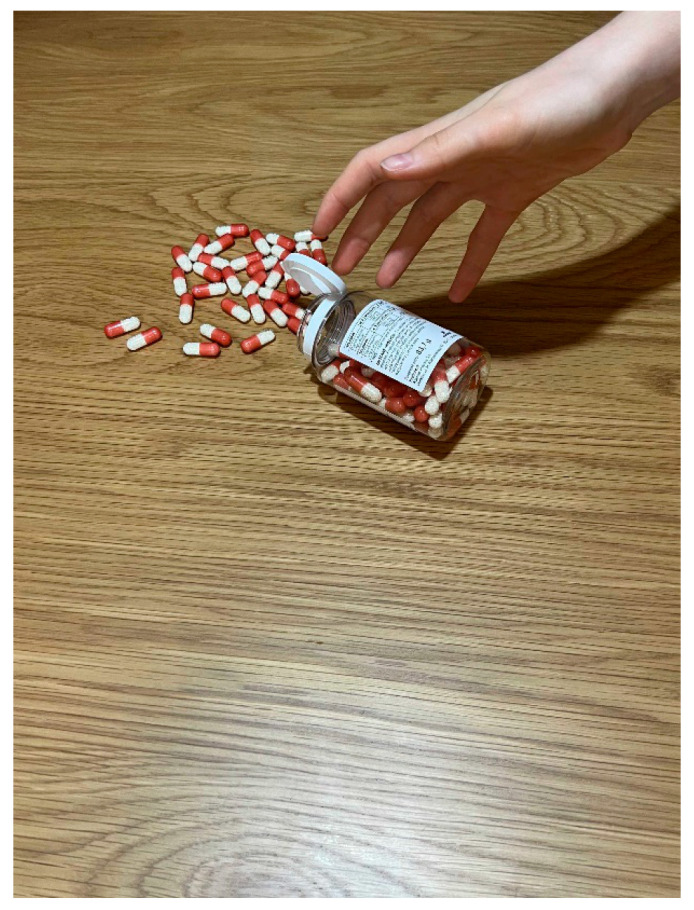


In the photo, we see a hand reaching toward red and white pills that have spilled out of a prescription bottle onto a wooden surface. The caption explains that aggression can cause the treatment process to be interrupted.

Now, let us look at a medical student’s symbolic photo. Figure 5 shows a healthcare workplace uniform and stethoscope around the collar. The uniform and stethoscope appear to symbolise a student of medicine on their rounds. The uniform is covered with sticky notes with writing on them.
Figure 5Sample symbolic photo and caption by a medical student participating in the photovoice activity. These are consequences of aggression: fear, lack of communication, fear of the patient disregard, lack of trust, low work comfort.
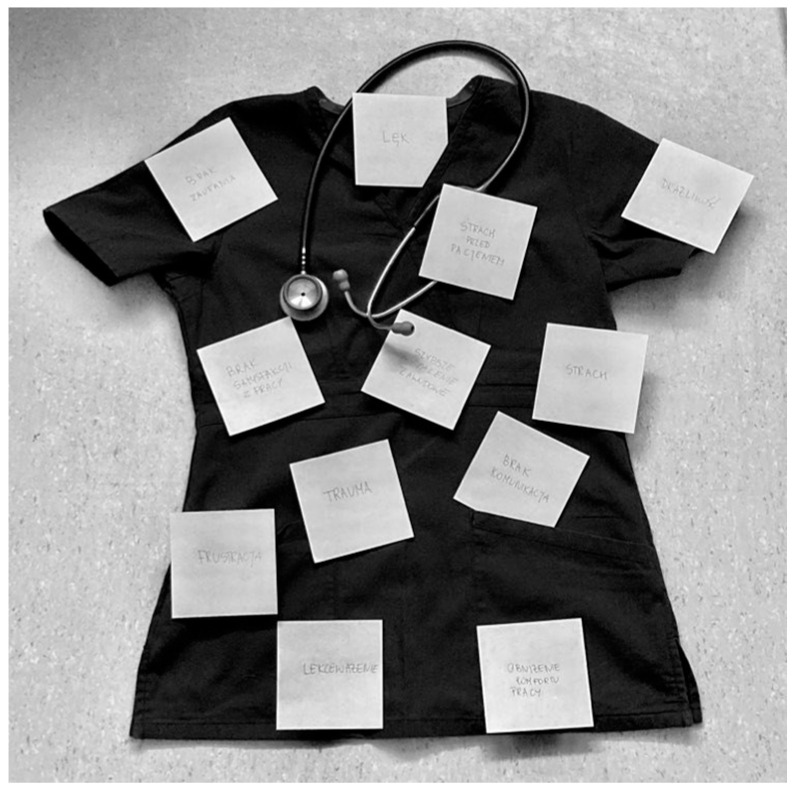


Words that are written on the sticky notes shown in Figure 5 photo are included in the caption. They describe the consequences of patient aggression in the workplace for the clinician (medical student).

Figure 6 is an example of a series of photos submitted by a medical student. The three photos show first a capped bottle standing up, then on its side with the cap off, and then broken into many pieces with the cap nowhere in sight.
Figure 6Sample photo series with a caption by a medical student participating in the photovoice activity. When you start working you are complete, but after aggression, you are broken.
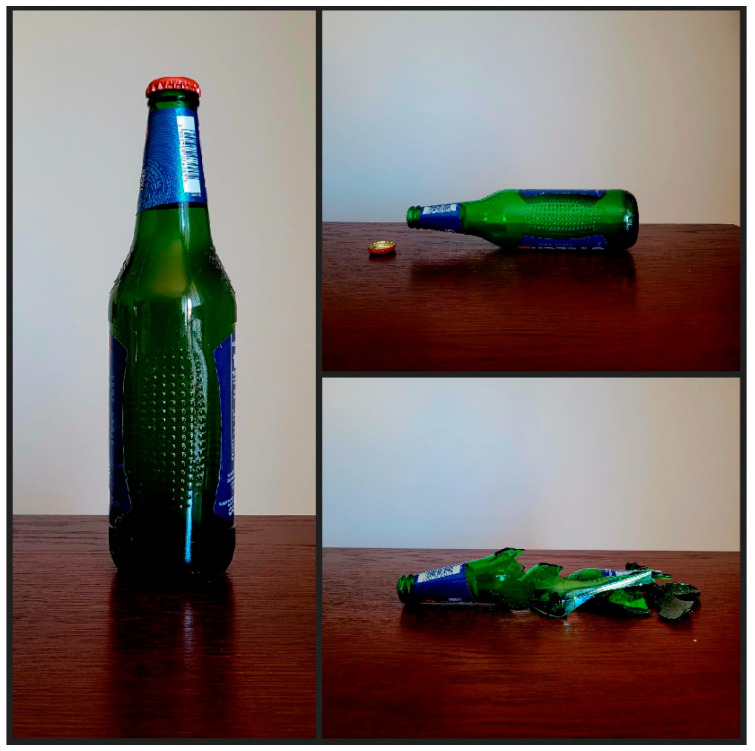


The photo series in Figure 6 shows the consequences of aggression on a student. The green bottle symbolises the student. As noted in the caption, the student feels broken after an incident of patient aggression.

## 4. Discussion

Students’ photos showed their perspectives on workplace aggression, and their photos created opportunities for insights into common workplace scenarios of aggression in healthcare. Students used the photovoice activity to show actual and imagined experiences and their impact on patients and themselves. The activity, which was a part of a 30-week AMP training, created an opportunity for students to be experts and to document their individual experiences and perspectives on the problem of aggression in the psychiatric workplace. Our study adds to the literature on the use of photovoice with students of health, nursing, medicine, public health, and health communication [15,16,17,18,19].

In our study, students of both disciplines, medicine and nursing, used the photovoice activity to take documentary and symbolic photos, which are common photo-taking strategies in photo-elicitation and photovoice projects [20]. We suggest that the open-ended photography assignment, which asked each student to submit one to three photos and would not be graded, encouraged students to be creative, reflect on aggression in the workplace and its consequences, and share perspectives that might otherwise have remained hidden. Facilitated discussion of the photos during a course session encouraged further reflection by the group before each student photographer explained the meaning of their photo or photos from their perspective. As in Andina-Diaz’s research, the photos stimulated students’ critical thinking and allowed them to look through the problem from a different perspective [17]. Haffejee (2021) noted that the photovoice activity moved students from passive to active learners and critical thinkers [15].

In their photos, students depicted well-known scenarios of aggression in the healthcare workplace. The images show actual situations as well as situations that could take place in the future. In our study, students of both disciplines took photos that illustrated evolving scenarios, such as aggressive behaviours affecting others or the impact of aggression on a student. Photovoice has also been used to prepare students for in-service learning, prior to immersion in an international context [19] (Ryan, 2020).

In our study, students from both disciplines took documentary photos, symbolic photos, and photos in a series. Prior studies using a photovoice activity as a part of the learning process have not described students’ photo-taking production strategies, as recommended by Rose (2023) [23]. In our study, medical students more often used a symbolic approach to photo-taking, which appeared to support the discussion of feelings about aggression and its impact in the workplace. Nursing students tended to use a more practical or documentary approach to depicting aggression in their training and workplace as experienced during their nursing training. Their photos often showed workplace tools to control aggression and facilitated discussions about difficult and sensitive issues, such as the use of coercive measures. Photos submitted by nursing students appear to align with quantitative research findings on the widespread aggression toward nurses in Polish hospitals, psychiatric wards, and primary care facilities [7], and suggest the imperative of requiring nursing student participation in the university’s AMP training. We recommend that future studies using photovoice to enhance student learning in health-related disciplines could explore and describe photo production more fully, to inform faculty decisionmaking related to instructional strategies.

During the photovoice activity, the students had an opportunity to show their personal standpoint and their individual views and perspectives on the problem of aggression in an inpatient psychiatric setting. The photographs show what students remember and imagine when they are asked to take photos. For the purposes of this pilot activity, the aim of increasing student engagement in their AMP learning was met, and engagement in learning is both a consistent goal and finding (both observed and measured) across studies using photovoice as a teaching strategy in health [18,24] (John & Samson-Akpan, 2021; Schell et al., 2009). As was the case in previous studies, the pictures allowed students to direct their attention to topics that had previously eluded them [16]. Future studies using photovoice with students could do more to interpret the photos, the actions portrayed, and the students’ standpoints to inform potential adjustments to the AMP curriculum or teaching methods.

The photovoice activity appeared to impact nursing students’ perceptions of aggression in the workplace. After the training, women saw aggressive behaviour as a defence of territory and thus a behaviour that served a particular purpose for the patient. The changing perception may have resulted from a better understanding of the motivation behind and causes of aggression. No such effect was observed in the male group.

Faculty observation indicates that, despite their initial negativity about the assignment, photovoice had a positive impact on medical students. In Poland, medical training focuses on technical skills, rote learning, and knowledge, not on critical thinking. Medical students were initially resistant to the training: “We were very angry because we had to do something unusual and difficult and because we did not have any guidelines, a scheme that fits”. After the activity they said, “That was interesting, productive, and made me think about the patient differently”. The positive emotions connected to the photovoice activity is very common and appeared in previous studies [15,16].

As seen in the photos taken by students in this pilot study, incorporating a visual medium and different forms of writing may stimulate students’ creativity and imagination and may meet the needs of diverse learners [19]. With the photovoice activity, students moved from passive to active learners and critical learners [15]. The assignment required active engagement in learning [16] and helped students to remember the course material (Haffejee, 2021) and their field observations (Dongre, 2011) [15,16]. Students appeared to focus more on the connection between their learning and their emotional or moral experiences when using photovoice when compared to traditional student reflection [18].

Two strengths of this pilot activity from our perspectives are the sample of students from two disciplines and the open-ended nature of the assignment. Expanding the pilot activity to include students from other healthcare disciplines could further inform the AMP curriculum and its methods. Future studies could explore whether students who use documentary versus symbolic photo-taking strategies or who take photos in a series show differences in outcomes related to critical thinking as compared to their peers. Longitudinal studies that follow students’ careers over time could explore whether incorporating a self-directed photography activity into AMP training to discuss the emotional impact of aggression in the healthcare workplace can have a longer-term impact on job satisfaction and retention. Two weaknesses are the small number of participating students and the lack of quantitative measures to track outcomes of interest. Using the photovoice method is not only attractive but also a cost-effective tool to facilitate critical thinking, develop research skills, and strengthen ties with colleagues [17].

In future, research could investigate whether enhancing the TERMA training with photovoice contributes to emotional learning and changes in attitudes toward aggression and aggressive patients. Pre/post questionnaires could be used to study changes, if any, in student empathy, attitudes, or critical thinking. Given the quantitative studies of AMP impact on students, incorporating quantitative measures in future studies using the photovoice activity in AMP training could help to decipher any changes in AMP effectiveness due to the use of a self-directed visual learning activity.

## 5. Conclusions

Adding photovoice to the AMP training appeared to be a valuable contribution to student reflection on their individual perspective on aggression in the workplace. The photos show how students experience incidents of aggression in the workplace. They used the photo assignment to visualise their fears in the workplace, their daily practical work, and their thoughts about future strategies to prevent or deal with violence in the workplace. Acting as student photographers during the TERMA training appeared to help them link their individual subjective experiences and their theoretical learning through the facilitated discussion of their photos, experiences, and feelings. Adding a photovoice activity to TERMA training created an opportunity to support student’s emotional engagement with their AMP learning. Our experience with this pilot suggests that enhancing TERMA training with a photovoice activity may positively influence student compassion for psychiatric patients and may strengthen their emotional equilibrium. However, caution must be taken when considering the impact of a photovoice activity with AMP students. Our study findings suggest that a photovoice activity is one way to surface the emotional impact of aggression in the psychiatric workplace and can be a useful technique for creating new understanding and insights and to initiate discussions. Future research could incorporate pre/post measures to decipher any differences in AMP student outcomes due to the addition of a photovoice activity.

## Data Availability

The data might be made available after direct contact with the corresponding author.

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
