# Peer review of "Photovoice in Aggression Management Training for Medical and Nursing Students—A Pilot Study"

_healthcare, 2024, doi:10.3390/healthcare12090873_

Round 1
Reviewer 1 Report
Comments and Suggestions for Authors
The topic is very interesting and the manuscript is well-structured. I just have some questions regarding the methodology of the study:
When recruiting students to participate, how did you avoid the appearance of coercion?
How did you allay students’ fear of retaliation if they did not participate in your study?
Also was the researcher the students’ teacher? If so, what potential biases and limitations were introduced?
Comments on the Quality of English Language
Minor editing of English language required
Author Response
| When recruiting students to participate, how did you avoid the appearance of coercion? | The Photovoice exercise was offered to the Students at the beginning of the course. They were informed about general rules, including their ethical aspects. They had the opportunity (as a group) to refuse the Photovoice activity as a part of their training without any consequences for their grade. However, the Students sees the activity as attractive and interesting, so they haven't felt any pressure by doing it. |
| How did you allay students’ fear of retaliation if they did not participate in your study? | The Students were asked to upload their pictures to the Uni-protected server, where only group members could see them. They were informed that at any stage, they could delete their photos from the server without any consequences or explanation to the teacher. |
| Also was the researcher the students’ teacher? If so, what potential biases and limitations were introduced? | To avoid biases, JL was a teacher and person responsible for collecting the data. He anonymised the data by giving them numbers so the rest of the Team could not identify them. LL and BK were responsible for picture analysis. |
Reviewer 2 Report
Comments and Suggestions for Authors
Manuscript is a clear, relevant fo the field a presented in a well-being manner. The manuscript has all part for research article according the Guide for Authors. The question is a original and well-being deefined. The manuscript is written in an approprate way (quality is high).
The part Materials and Methods is original and relevant. The methodology is clearly and comprehensibly described. The ethics statements and data availability statements are adequate (we can find it in Materials and Methods and at the end of the text of manuscript, before part References). But iIt is necessary to add information about the duration of the study (when the study took place) to the Material and Method sections of the manuscript.
The results are interpreted appropriately.
Discussion is written in appropriate way.
Conclusions are consistent with the evidence and arguments presented and are presented clearly and comprehensibly. The conclusions are interesting for the readership of the journal.
All references are appropriate. Cited references are recent publications and mostly recent publication within the last 5 years. References does not include the self-citations. There are little mistakes in the notation of literature in the references, you need to check each source and unify the notation, such as dots, colons, commas.
All (six) figure in the manusript are appropriate and they properly show the data. Marking of the description of the table Figure 5 Sample....... - should be in bold font. Below each figure is is decription of its content in understandable way. The content of figures are interpreted appropriately and consistently throughout the discussion of manuscript.
Please, fill in the data for the surname under number 2.....in the affiliation section (city, country)
Overall recommendation for next processing stage of the manuscript: accept after minor revisions.
Author Response
| Manuscript is a clear, relevant fo the field a presented in a well-being manner. The manuscript has all part for research article according the Guide for Authors. The question is a original and well-being deefined. The manuscript is written in an approprate way (quality is high). | |
| The part Materials and Methods is original and relevant. The methodology is clearly and comprehensibly described. The ethics statements and data availability statements are adequate (we can find it in Materials and Methods and at the end of the text of manuscript, before part References). But iIt is necessary to add information about the duration of the study (when the study took place) to the Material and Method sections of the manuscript. | We have added further information on using photovoice as a teaching tool (immediately before materials and methods). We have also added content and citations related to analyzing photovoice data (visuals and text) to the methods section. The study took place in 2023. |
| The results are interpreted appropriately. | We hope that the new description of our analysis methods will support the credibility of our findings, which are descriptive, which is common with photovoice studies to date. |
| Discussion is written in appropriate way. | We have added further content and citations throughout the discussion section. |
| Conclusions are consistent with the evidence and arguments presented and are presented clearly and comprehensibly. The conclusions are interesting for the readership of the journal. | |
| All references are appropriate. Cited references are recent publications and mostly recent publication within the last 5 years. References does not include the self-citations. There are little mistakes in the notation of literature in the references, you need to check each source and unify the notation, such as dots, colons, commas. | The citations were checked and corrected. We have added further citations. |
| All (six) figure in the manusript are appropriate and they properly show the data. | |
| Marking of the description of the table Figure 5 Sample....... - should be in bold font. | done |
| Below each figure is is decription of its content in understandable way. The content of figures are interpreted appropriately and consistently throughout the discussion of manuscript. | Thank you for this comment, and in the methods section we have included content on our approach to analysis, particularly descriptive analysis of photos. |
| Please, fill in the data for the surname under number 2.....in the affiliation section (city, country) | The information is added to the text |
Reviewer 3 Report
Comments and Suggestions for Authors
The topic is original and provides a way to empower healthcare students to understand patient aggression. The article is well-written. The article depicts: "Students’ photos showed their perspectives on workplace aggression and created opportunities for insights into common workplace scenarios of aggression in healthcare" but not: "...treatment tools, and prevention strategies..." as indicated in the abstract. Because it is a pilot study, caution must be taken to make conclusions from the method as a way of helping students to manage patient aggression. It seems to be more of a method to create understanding and insights and initiate discussions on the topic.
Line 84: sentence repeated.
The methods require a better explanation of how the data generated from the photos will be analysed to reach conclusions. Some conclusions were drawn from how nursing and medical students differed in their approach to taking the photos. I am not sure that such interpretations can be made without scientifically analysing the photos following measures to ensure trustworthiness.
The discussion needs more literature integration as only a few resources were used.
Author Response
| The topic is original and provides a way to empower healthcare students to understand patient aggression. The article is well-written. The article depicts: "Students’ photos showed their perspectives on workplace aggression and created opportunities for insights into common workplace scenarios of aggression in healthcare" but not: "...treatment tools, and prevention strategies..." as indicated in the abstract. | We've deleted this part of statement, as suggested. |
| Because it is a pilot study, caution must be taken to make conclusions from the method as a way of helping students to manage patient aggression. It seems to be more of a method to create understanding and insights and initiate discussions on the topic. | we've added this valuable comment to the Conclusion section |
| Line 84: sentence repeated. | The repeated phrase is deleted. |
| The methods require a better explanation of how the data generated from the photos will be analysed to reach conclusions. Some conclusions were drawn from how nursing and medical students differed in their approach to taking the photos. I am not sure that such interpretations can be made without scientifically analysing the photos following measures to ensure trustworthiness. | our research methodology |
| The discussion needs more literature integration as only a few resources were used. | We have added new content and citations to the discussion section. |